

# Mechanically-driven stem cell separation in tissues caused by proliferating daughter cells

Johannes C. Krämer[1], Edouard Hannezo[2], Gerhard Gompper[1] and Jens Elgeti[1⋆]

**1** Theoretical Physics of Living Matter, Institute of Biological Information Processing and Institute for Advanced Simulations, Forschungszentrum Jülich, 52425 Jülich, Germany
**2** Institute of Science and Technology Austria, 3400 Klosterneuburg, Austria

⋆ j.elgeti@fz-juelich.de

## Abstract

The homeostasis of epithelial tissue relies on a balance between the self-renewal of stem cell populations, cellular differentiation, and loss. Although this balance needs to be tightly regulated to avoid pathologies, such as tumor growth, the regulatory mechanisms, both cell-intrinsic and collective, which ensure tissue steady-state are still poorly understood. Here, we develop a computational model that incorporates basic assumptions of stem cell renewal into distinct populations and mechanical interactions between cells. We find that the model generates unexpected dynamic features: stem cells repel each other in the bulk tissue and are thus found rather isolated, as in a number of in vivo contexts. By mapping the system onto a gas of passive Brownian particles with effective repulsive interactions, that arise from the generated flows of differentiated cells, we show that we can quantitatively describe such stem cell distribution in tissues. The interaction potential between a pair of stem cells decays exponentially with a characteristic length that spans several cell sizes, corresponding to the volume of cells generated per stem cell division. Our findings may help understanding the dynamics of normal and cancerous epithelial tissues.

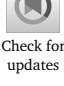

## Contents



# 1  Introduction

Tissue renewal through cell division to balance the constant loss of cells is a hallmark of life in multicellular organisms. It is widely accepted that for most tissue types, stem cells (SC) play a key role in this complex process [1,2]. Stem cells have the potential to proliferate and self renew over long timescales, continuously generating various types of differentiated cells required for physiological tissue function. Such tissue maintenance via small populations of stem cells requires fine-tuned fate choices to ensure not only a constant and well-defined ratio of cellular types, but also stable tissue size and overall cellular numbers.

Equally important, however, is a homogeneous positioning of stem cells across the tissue, such that lost cells can quickly be replenished. If all stem cells were to segregate from progeny, tissue function and maintenance could be compromised. Indeed, in a lattice model without mechanical interactions, proliferating and non-proliferating cells have been shown to require reversible differentiation in a manner dependent on relative local cell numbers to avoid non-homeostatic behavior, such as phase separation of cell types [3]. Other types of fate control that have been recently explored theoretically and experimentally are an effective fate determinant field of a diffusible chemical. It is homogeneously supplied but consumed by stem cells, thus, providing a biochemical negative-feedback mechanism for stem cell density, which can also reproduce homogeneous distribution of stem cells within the tissue [4].

In recent years, mechanical forces have been shown to be crucial regulators for tissue growth and homeostasis. A key concept is the *homeostatic pressure*, which is the pressure at which cellular division and apoptosis are balanced, under the generic assumption that these processes depend on mechanical forces. Based on this concept, competition between different tissues [5], tissue fluidization [6], negative bulk homeostatic pressure [7], interface dynamics during competition [8–10], and the evolution of tissue [11] could be explained. However, these studies always considered situations without the possibility of conversion between cellular types and change in self-renewal potential, which is by definition key to understand stem

(a)

(b)

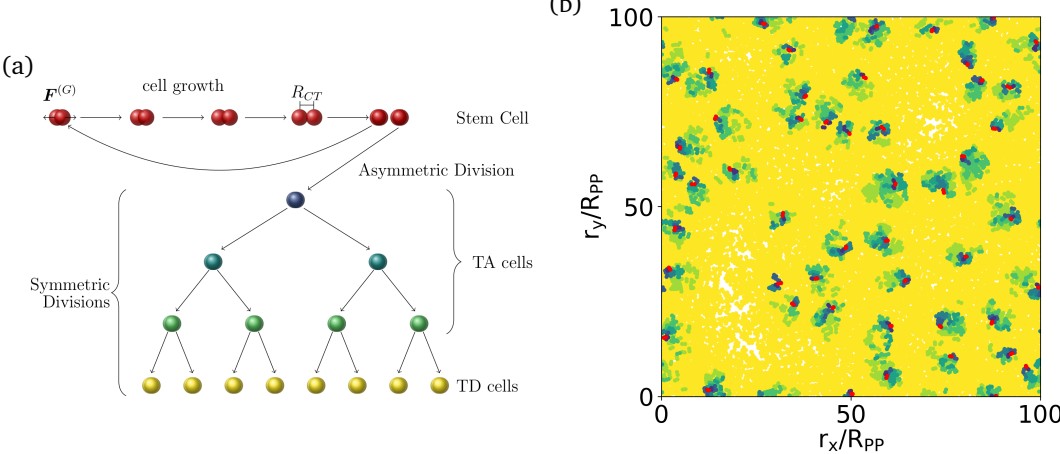

Figure 1: **Stem cell (SC) division and differentiation scheme leads to well-separated SC arrangement.** (a) Schematic illustration of growth, division, and differentiation implemented in the two-particle growth model. SCs grow by an active growth force $F^{(G)}$ and divide when the two particles of a cell are separated by a threshold distance $R_{ct}$. A SC divides always asymmetrically in one SC daughter and one TA daughter cell. TA cells still grow, but their maximal number of divisions are limited. They always divide symmetrically, and in the final step both daughters are TD cells. TA and TD cells are removed with the same, finite apoptosis rate $k_a$. (b) In bulk simulations, SCs are distributed all over the tissue, and rarely at close distance. Each SC is surrounded by a halo of TA and TD cells. The main mass is made up by non-growing TD cells. The same color scheme as in Fig. 1a is applied with finer gradation for TA cells, which can divide up to five times. The simulation was performed in a squared two-dimensional box with box length 100 $R_{PP}$. After equilibration, the average cell density is $\rho = 1.58/R_{PP}$, and the snapshot was taken after 25 generations.

cell dynamics in tissues. This raises the question of whether purely mechanical interactions, which must occur in any confluent tissue with cell renewal and loss, could be a factor in the regulation of stem cell dynamics and positioning.

Here, we study the interaction and distribution of stem cells in self-renewing tissues in the presence of mechanical interactions only. As a minimal hierarchical scheme of tissues with stem and differentiated cells (Fig. 1a), we take a classical model [12], where stem cells (SC) asymmetrically divide into a stem and a transient amplifying (TA) cell. The TA cells can undergo a number $N_{TA}$ of symmetric divisions before terminally differentiating into non-proliferative cells (TD). Although this is an oversimplified picture [13], we show that it already gives rise to complex dynamics and provides a computational framework which can be readily generalized to more realistic stochastic models of cell differentiation. This includes, for instance, effects of the stem cell environment, the so called niche, on fate decisions [14,15] or self-controlled fate decisions such as in "open niche" [16], where all cells have the same differentiation potential. Implementing the renewal scheme into the two particle growth model [17] (sec. 2 and App. B), we find that the resulting tissue is characterized by well separated SCs surrounded by a small cloud of TA cells in a sea of TD cells in two (see Fig. 1b), and three dimensions (see sec. 6). The cause of the homogeneous distribution of stem cells in the tissue on larger distances lies in the of outflux of cells from the stem cells, and short-range mechanical interactions. Surprisingly, we find that this repulsion and distribution of SC in the tissue can be effectively described by a simple thermodynamic picture with SCs behaving like soft

repulsive colloids in a thermal bath. The length scale of the soft repulsive potential is set by the volume of cells generated per stem cell division.

In section 3, we first discuss the dynamics of single stem cells to determine the length scale of the interaction, and to study how the cloud of their progeny around them impacts their movements. Subsequently, in section 4, we study the dynamics of SC pairs. From their distance distribution function, we infer the amplitude of the effective repulsive potential that is mediated by the outward flow of differentiated cells that they each produce. In section 5, we show that the distribution of stem cells in a bulk tissue can be described by this effective pair interaction. In three dimensions, we find that the repulsive interaction of stem cells is also preserved (see sec. 6). Last, we demonstrate independence of the homeostatic pressure controlled division mechanism, by replacing it with a stochastic one, in section 7. This shows that our results are primarily driven by mechanical interactions and proliferation generated outflow of cells.

## 2   Simulation model

Each cell in the simulation model consists of two particles, which interact via a repulsive force to model growth. When the particle distance reaches a size threshold, a cell divides and two new particles are placed at random at very short distance next to the old ones. Particles of different cells interact via an attractive and a repulsive force, to model adhesion and volume exclusion, respectively. The length $R_{PP}$, up to which inter-cellular interactions are present, defines the length scale of our model and depicts the cell size. TA and TD cell removal is modeled in a stochastic process with fixed rate $k_a$, which defines the timescale of the model. Note that the division rate $k_d$ is not fixed and will follow from the other model parameters. Random and dissipative forces are included by dissipative particle thermostat. Together with a self-consistent velocity Verlet integrator for the Langevin equations, this enables efficient calculations, correctly accounting for the overdamped dynamics required for tissue simulations, as discussed in detail in App. D.3. Cells follow a classical differentiation model [12], see Fig. 1a, and all have identical interaction parameters. The only difference between them is that stem cells do not die ($k_a = 0$), and TD cells do not actively grow ($G = 0$) to prevent divisions. All simulations in this work are initialized with SCs only. More details on the model can be found in App. B and previous works [7, 8, 10, 11, 17].

## 3   Isolated stem cell

To understand the separation of stem cells in bulk tissue, we begin our analysis by studying the dynamics of a single SC and its offspring. We start with an isolated stem cell and shortly thereafter find it surrounded by a cloud of progeny that maintains a certain size throughout the simulation time (see Fig. 2a).

Assuming equal division rates $k_d$ for SC and TA cells and neglecting TA cell loss on average $2^{N_{TA}}$ growing cells are present at a time, with $N_{TA}$ the maximum allowed number of TA cell division cycles. Further, $2^{N_{TA}}$ TD cells are produced in each cycle. Thus, each stem cell division can produce at most $2^{N_{TA}+1}$ progeny. However, due to the stochastic loss rate $k_a$ of TA and TD cells in our model (which can occur either due to apoptosis in monolayers, or out-of-plane delamination in multilayered tissues such as the epidermis) we arrive at a lower effective number $\langle N_p \rangle < 2^{N_{TA}+1}$ (see Fig. 2c). On average, the stem cell is thus surrounded by a circular arrangement of its offspring, with a characteristic length $L_p = \sqrt{\langle N_p \rangle / \rho \pi}$, where $\rho$ denotes the cell density (see Fig. 2d). Note that $k_d$ is not fixed in our simulation model, but controlled

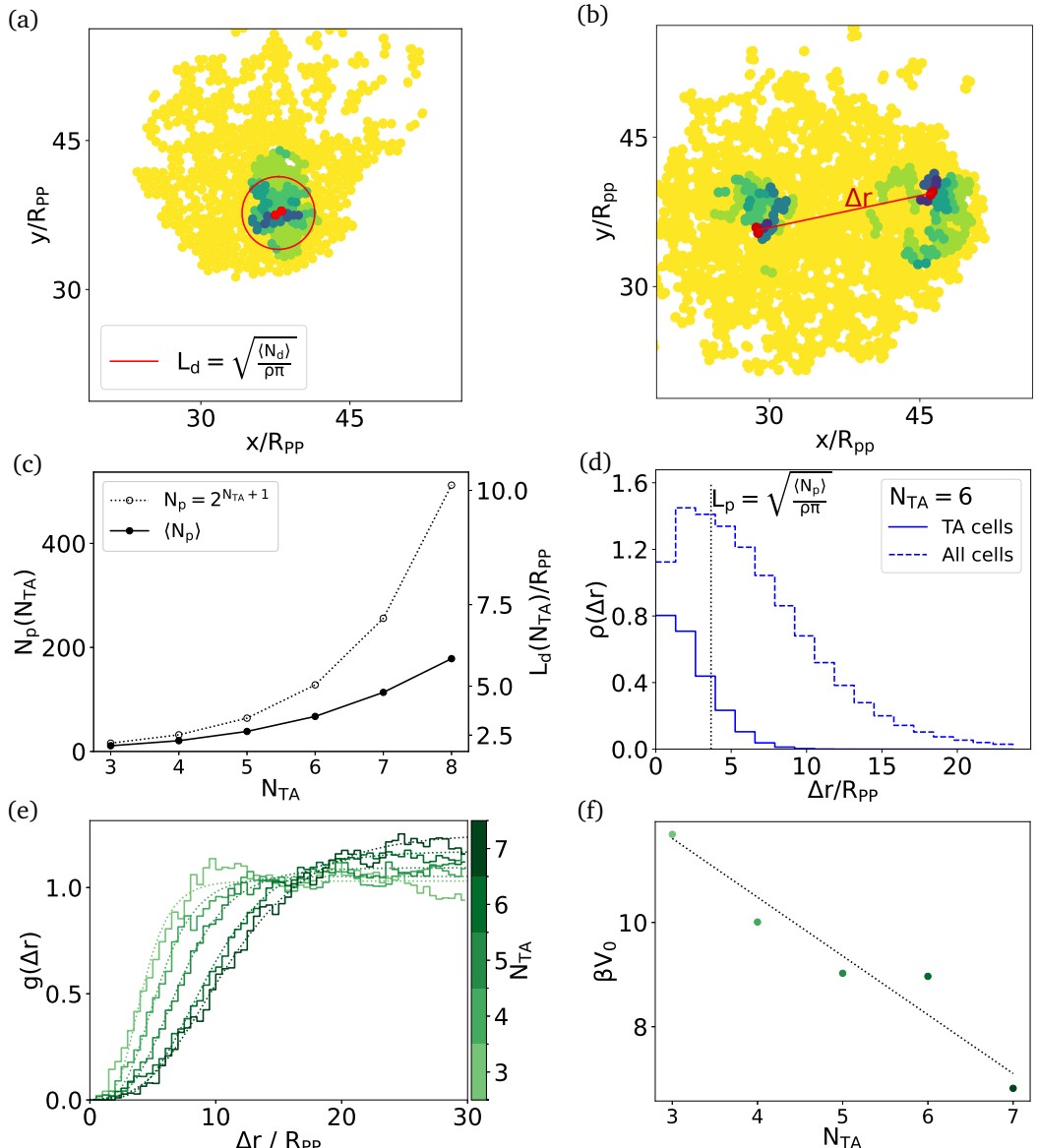

Figure 2: **Interaction parameter determination.** (a) Snapshot of isolated SC and (b) two interacting SCs. (c) Cell number generated in each SC division cycle, i. e. sum of TA cell and latest TD cells, for the upper limit $N_p = 2^{N_{TA}+1}$ (open circles, dotted line) and measured from simulation by averaging over time $\langle N_p \rangle$ (filled circles, solid line) as function of $N_{TA}$. The latter is used to determine the characteristic progeny distance $L_p$, which is used on the right y-axis label. (d) Cell density of TA (solid line) and all daughter cells (dashed line) around SC, where $L_p$ derived from average measured progeny number $N_p$ is marked by dotted line, for $N_{TA} = 6$. (e) Pair correlation $g(\Delta r)$ as function of SC distance $\Delta r$ for a pair of SCs. $N_{TA}$ is coded in color. Fit shows interaction via repulsive potential and obtained interaction strength $\beta V_0$ is shown in (f) as a function of $N_{TA}$. Multiple independent long simulation runs of these systems for each $N_{TA}$ are used to determine the interaction parameters. Simulations are initialized from a single SC (a, c, d) and two SCs in contact (b, e, f) and averages are taken in steady state, i. e. after reasonably long time after initialisation.

by pressure [5,7,18]. An increase of TA cell generations can thus decreases the proliferation rate and effective cell number further (see App. C Fig. 7).

However, the average hides an important aspect of the dynamics. While in some instances the SC is indeed in the center of the cell mass, it is also often found at the boundary due to the stochastic nature of the orientation of asymmetric divisions. When the SC is located at the boundary, the divisions of the TA cells result in an effective propulsion of the SC and persistent directed motion (App. D Fig. 8).

Correspondingly, the mean squared displacement (MSD) of the stem cell shows a regime of active motion up to about the lifetime of one generation, and a regime of activity enhanced long-term diffusion thereafter (see Fig. 4a and App. D Fig. 9). In analogy to active Brownian particles (ABPs) or Run'n'Tumble particles, the MSD of stem cells can be described in terms of a translation diffusion coefficient $D_t$, a characteristic rotational time $\tau_R$, which confirms decorrelation of the SC motion after about one cell generation, and a propulsion velocity $v_0$ (see App. D.1). The fitted values for $\tau_R$ are about one half to one apoptosis times and correspond to the decorrelation of directed motion. For the propulsion velocity we find a maximum for $N_{TA} = 6$, indicating an optimal population size for sped-up motion.

We quantify the active outbursts of SC motion by the displacement distribution of the stem cell, i.e. the probability that the SC undergoes a given displacement within a fraction of a generation (here $0.2/k_a$ see App. D Fig. 11a). Heavy tails in the distribution display the persistent motion during outburst. When a stem cell is displaced from the center of mass of the cell population the heaviness of the tail increases strongly (see App. D Fig. 11b).

## 4 Stem cell interaction

In order to derive an effective pair interaction, we initialize the simulation with two stem cells in a large periodic box. Each stem cell generates its own progeny populations, which form an aggregate via which stem cells effectively repel each other (see Fig. 2b). Loss of cells in the region between stem cells can lead to two separate cell populations, both behaving as described in the previous section. In simulations with periodic boundary conditions, they eventually collide again and SC repulsion can be observed yet again. We quantify the stem cell repulsion by measuring the pair correlation function

$$g(r) = \left\langle \sum_{m,n \neq m}^{N_{SC}} \Theta_H (r - r_{mn}(t_i)) \Theta_H (r_{mn}(t_i) - (r + \delta r))/\Omega(r) \right\rangle ,$$

in an annulus ranging from $r$ to $r + \delta r$, where $\Theta_H(x)$ is the Heaviside function and $r_{mn}(t_i) = |\boldsymbol{r}_m(t_i) - \boldsymbol{r}_n(t_i)|$ the stem cell pair distance, measured applying minimum image convention. The average combines data from different times and independent runs to improve statistics and the sum runs over pairs of SC. In order to normalize $g(r)$, and to take into account the radial bin size, we divide by a geometrical scaling constant $\Omega(r) = L_{XY}^2/[4(N_{SC}-1)((r+\delta r)^2 - r^2)]$. The probability to find two SCs at short distances is strongly reduced, and reaches a plateau at larger distances, where the cell clouds are out of contact again (see Fig. 2e).

Next we utilize an analogy to thermal equilibrium to derive an effective repulsive pair potential $V(r) = V_0 \exp\{-r/L_{rep}\}$ of very soft colloids. The repulsion length $L_{rep}$ is set to the characteristic length $L_p$ of proliferating progeny around the SC. In equilibrium, the pair correlation function follows from Boltzmann statistics

$$G(r) = G_0 \exp\{-\beta V_0 \exp\{-r/L_p\}\} , \tag{1}$$

with $\beta = 1/k_B T_{eff}$ and effective temperature $T_{eff}$. The prefactor $G_0$ is obtained from the normalization condition $\int G(x)dx = 1$. We determine the effective potential strength $\beta V_0$

by fitting $G(r)$ to $g(r)$ and find good agreement of this simple model with our data. Results are shown in Figure 2e+2f. The decrease of $\beta V_0$ with $N_{TA}$ corresponds to a softening of the potential, which could be due to an increasingly asymmetric distribution of TA cells.

## 5   Confluent tissue maintained by stem cells

We perform tissue simulations for different $N_{TA}$, and vary the number of stem cells $N_{SC}$ in the simulation such that the average total cell density is approximately the same for different simulations. We compare these tissue simulation with Brownian dynamics (BD) simulations of thermal colloid particles interacting via the proposed effective repulsion. The interaction parameters, obtained in section 4, are used without further adaptions. Snapshots for the mechanical model and the thermal colloid model are shown in Figure 3a and 3b, respectively.

Clearly, SCs are well separated, and homogeneously distributed over the system. To quantify this separation, we measure again the pair correlation function for both systems (see Fig. 3c). The pair correlation function reveals a strong depletion of stem cells from the vicinity of another on length scales set by the amount of progenitors they generate. Interestingly, we find that the colloid system (dotted lines) reproduces well the results from the tissue simulations (solid lines) even in confluent tissues. Note that this requires a rather soft colloidal interaction on length scales much larger than the (stem) cell size.

We further quantify the stem cell separation with a cluster analysis. We consider that SCs which are found at a distance smaller than $d$ belong to the same cluster, and calculate the cluster size distribution function $\mathcal{N}(n) = p(n) \cdot n$, where $p(n)$ is the probability of a cluster of $n$ SCs, and average cluster size $\langle n \rangle$. If we take the distance threshold $d$ equal to the cells interaction range $R_{pp}$, as commonly done for cluster analysis, we identify close to no clusters at all ($\mathcal{N}(n=2) = 0.024$ and $\mathcal{N}(n=3) = 0.00014$, for $N_{TA} = 5$). By choosing a larger cutoff $d = 2L_p$ on the basis of their progeny, we find some small clusters, but still not any large clusters (see Fig. 3d and App. E Fig. 14), a feature indicative of SC separation, which is also reproduced in the colloidal systems.

To obtain more insight in the structural dynamics of both systems, we additionally calculate the MSD of SC, and the intermediate scattering function

$$S(q,t) = \left\langle \sum_{m=1}^{N_{SC}} \sum_{n=1}^{N_{SC}} e^{i\boldsymbol{q}\boldsymbol{r}_m(t_0+t)} e^{-i\boldsymbol{q}\boldsymbol{r}_n(t_0)} \right\rangle,$$

where $\langle \cdot \rangle$ denotes a time average over $t_0$ and we calculate the over $S(q,t)$ with wave vector $q = |\boldsymbol{q}|$ (see Fig. 4b).

The MSD of SCs in tissues shows the same superdiffusive behaviour as found for an isolated SC in free space (see Fig. 4a). However, the displacement is reduced in confluent tissue due to the increased cell density. We find, that this affects mainly the translational diffusion coefficient and propulsion velocity, while the reorientation remains in the same order of magnitude as for isolated SC (see App.D Fig. 10). The thermal colloid system on the other hand shows normal diffusion, as should be. Because the Boltzmann description does not provide any timescales, we use the short-term diffusive limit of colloids and SCs in tissues to fix the timescale. The short-term diffusion coefficient can also be calculated from the relaxation of the intermediate scattering function as $D_{eff} = -\log\{S(q,t)/S(q,0)\}/(q^2 t)$ in the short time limit. After aligning the time scale of the thermal colloid system, we find that this effective diffusion is approximately the same for both systems (see App. E Fig. 15). For large $q$, $D_{eff}$ is constant, while for small $q$, finite-size effects of the simulation area come into play.

For zero time lag ($S(q, \Delta t = 0)$), we find a plateau, corresponding to the autocorrelation term ($n = m$). With a finite time lag, the scattering function starts to decay with $q$ (see Fig. 4b

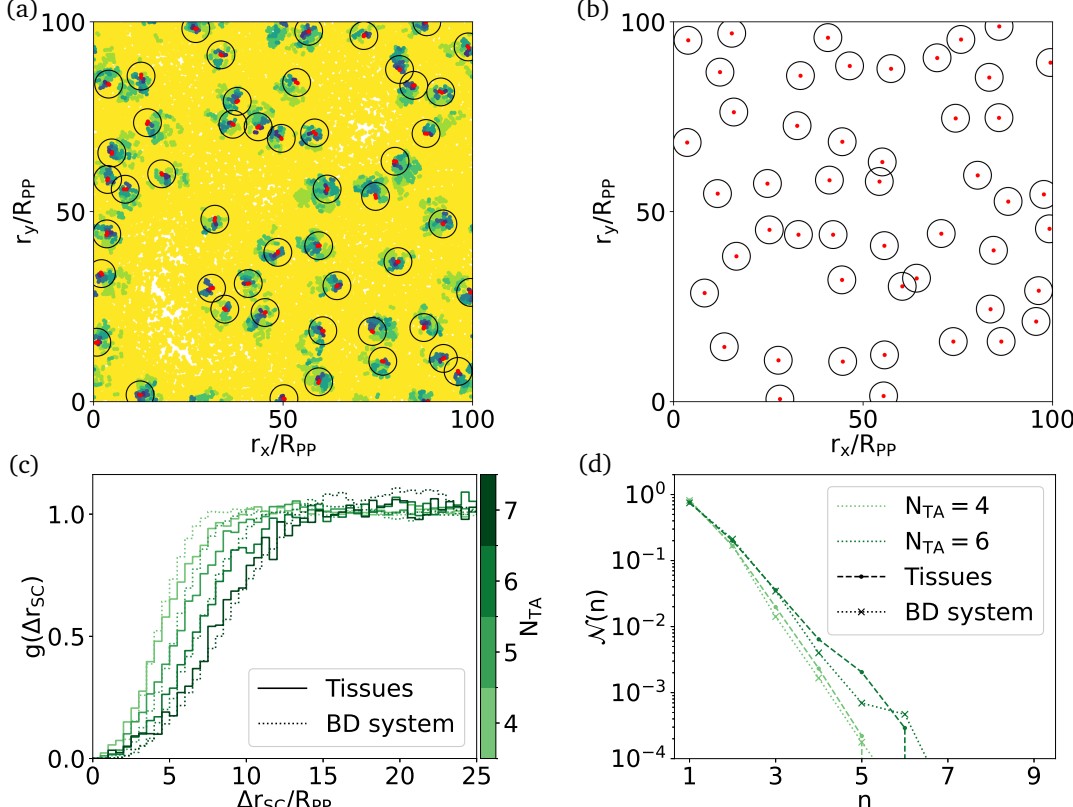

Figure 3: **SC separation in bulk tissue is explained by equilibrium model with simple repulsive interaction.** (a) Snapshot of the tissue simulation for $N_{TA} = 7$ using the color scheme introduced in Fig. 1 and (b) respective thermal colloid simulation. The characteristic progeny length scale is displayed by circles. (c) The SC and particle pair correlation functions, measured from simulations of the two-particle growth model (solid lines) and thermal colloids, interacting with the assumed repulsive interaction (dotted lines), are shown for color coded $N_{TA}$. (d) Cluster size distribution function for SCs in tissues (dots, dashed line) and thermal colloids (cross, dotted line) for color coded $N_{TA}$. Distance threshold for clustering is set to $d = 2L_p$, much larger than the cell size, in order to see any clustering at all. The number of SCs was chosen such that the cell density in all simulations is $\sim 1.58\,\text{cells}/R_{PP}$, and the number of thermal colloids equals the number of SCs. Interaction parameters of colloids were extracted from single and two SC simulations. Note the astonishing agreement between both systems. Simulations are initialized with fixed number of SCs or thermal colloids at regularly spaced distances and averages are taken in steady state, i. e. after reasonably long time after initialization.

for $\Delta t = 0.07k_a$). For tissues and colloids, we show the relaxation of small and large structures in Fig. 4c and Fig. 4d, respectively. Because we match the timescale of both systems for the short-term diffusion, we find that small structures relax at similar times. Still, SCs in tissue show faster relaxation compared to the BD system. For larger structures (small $q$), the SCs in tissue relax significantly faster compared to the thermal colloids due to their superdiffusive motion. The enhanced relaxation with increased structure size holds for all $q$ as can be seen from the half life relaxation time (see App. E Fig. 15).

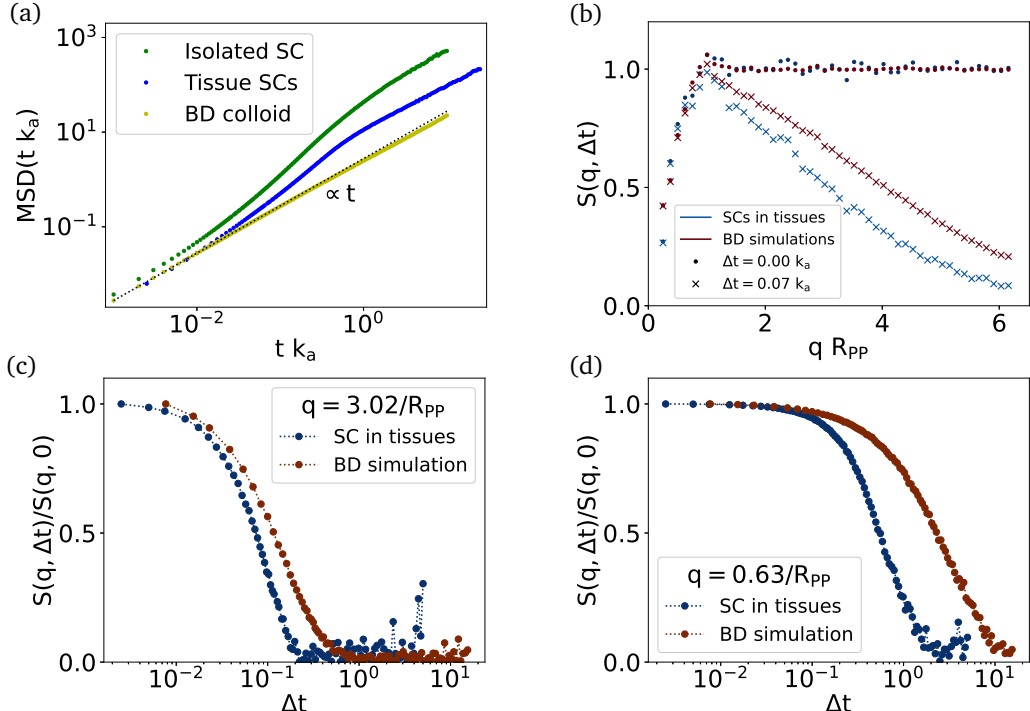

Figure 4: **Dynamics of SC and BD simulations.** (a) Mean squared displacement (MSD) for isolated SC (green) and SCs in tissue (blue), both with $N_{TA} = 4$, and respective thermal colloid system (yellow). In the tissue model, we find superdiffusive SCs. Their displacement gets reduced with increased cell density in confluent tissues. Time scale of thermal colloid system was aligned with the short term diffusion of SC in tissues. (b) Dynamic structure factor (DSF) as function of wavevector for $t = 0$ and $t = 0.07k_a$. Relaxation of the DSF as function of time for (c) small structures ($q = 3.08/R_{PP}$ or $\lambda_q = 2.03R_{PP}$, i.e. twice the cell diameter) and (d) large structures ($q = 0.69/R_{PP}$ or $\lambda_q = 9.11R_{PP}$, i.e. comparable to "weakly" interacting stem cells). Small structures relax at comparable timescale, whereas large structures relax faster in the tissue model due to the superdiffusive behavior of stem cells. Simulations are initialized with fixed number of SCs or thermal colloids at regularly spaced distances and averages are taken in steady state, i.e. after reasonably long time after initialization.

# 6 Stem cell repulsion in three dimensions

Many biological tissues are not restricted to two dimensions, but extend to the third dimension, which is especially the case in cancer, where cells grow out of plane and form spheroidal tumors. We show that the effect of repulsion by progeny is not limited to two dimensions and can also be found in tissue simulations in three dimensions (see Fig. 5). Qualitatively, the pair correlation function shows the same repulsion of SC at a distance corresponding to the volume of their progenitors. Note that here the geometric normalization of the pair correlation function is the volume of a sphere instead of the area of a circle around the stem cell.

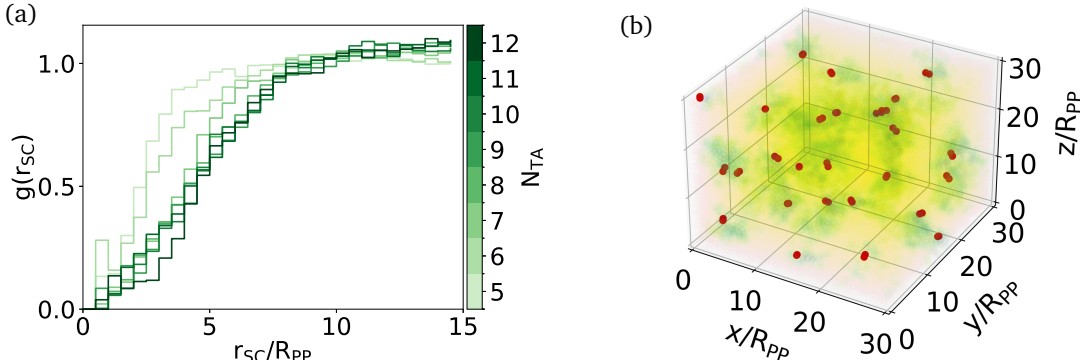

Figure 5: **Stem Cell repulsion qualitatively remains in three dimensions.** (a) Pair correlation function for stem cells in three dimensions shows reduced probability to find SCs at short distances, and plateau at longer distances. (b) Snapshot for a three dimensional tissue with $N_{TA} = 10$ after 50 generations, using the color scheme introduced in Fig. 1. All simulations are performed with 27 SCs in a cubic box with box length 30 $R_{PP}$.

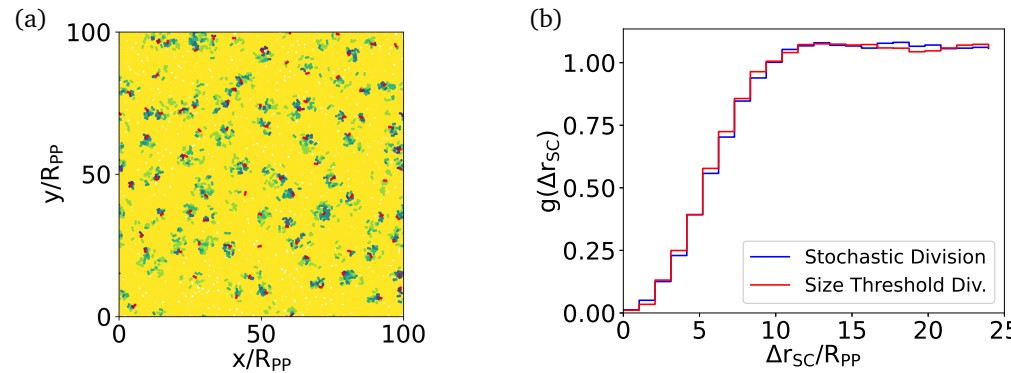

Figure 6: **Relevance of homeostatic pressure growth control.** (a) Snapshot for deterministic TA cell differentiation ($N_{TA} = 6$) in the model with stochastic division mechanism, using the color scheme introduced in Fig. 1. (b) Pair correlation function for simulation model with division after reaching a size threshold (red) and stochastic divisions (blue).

# 7 Independence of homeostatic pressure controlled divisions

Stem cells interacting effectively via their (growing) progeny is a robust effect in tissues with mechanical cell-cell interactions. We show this by replacing the division after reaching a size threshold, which introduces a feedback on the homeostatic pressure, by stochastic divisions with fixed rate $k_d$. This rate is set to the measured division rate in the size threshold division model for an exemplary tissue with $N_{TA} = 6$. Details on the implementation can be found in Appendix F.

The distribution of TA cells around the SC remains almost unchanged (see Fig. 6a), thus, also the SC pair correlation function of both models remains the same (see Fig. 6b). The SC dynamics, measured by the MSD, remains unchanged as well (see App. F Fig. 17). Thus, the mechanical cell-cell interactions in the model and outflux of cells from the SC, rather than the feedback of growth and division on pressure, are generating the effective SC repulsion.

# 8   Discussion and conclusion

We have studied the dynamics of stem cells in self-renewing tissues driven by mechanical interactions. Due to the constant outward flow of progeny generated by stem cells, we find that SCs effectively repel each other and spontaneously organize into dispersed structures with a corona of their progeny. This provides an updated and more biophysical view of the classical concept of proliferating units in tissues, in which size and shape of self-renewing domains are flexible, but still organized by stem cells which are robustly present within each unit [19].

Interestingly, we find that the interaction between stem cells can be largely described in a simplified fashion with help of an effective equilibrium approach. The assumption of a purely repulsive interaction and determination of the interaction length from the spatial requirements for the progeny produced by each stem cell division allows us to characterize SCs like very soft colloids, which interact via this effective repulsive interaction – much larger than the cell size. Further, we have shown that this simplified model reproduces the structural results of the two-particle growth model in confluent tissues. The pair correlation function of the thermal colloids is remarkably similar to that of actively growing cells. The dynamics however are different. We show evidence of an active and persistent self-propulsion force acting on SCs due to the surrounding growing cells. We systematically characterize these, in particular in terms of the statistics and active nature of their persistent random walks. As a result of this proliferation-driven movements, the confluent tissue shows faster relaxation characteristics on long time and length scales. The only detectable difference between the actively growing tissue and the thermal colloid system was found in the dynamic structure factor. It shows a speedup of large structure relaxations in the tissue model due to its active nature, and thus confirms that the active system can be mapped surprisingly well to an thermal equilibrium system.

The derivation of an effective interaction using nearest neighbor distribution functions and mapping it on particle systems may lack sensitivity to certain characteristics of the interaction, especially with increasing density [20, 21]. However, they are still able to capture the main influences of the interaction. Further, replacing the passive Brownian particle system by an active one, is very likely to fail as the continuously decrease of activity enhanced motion cannot be derived from first principles of infinitely diluted systems [22].

Small populations of SCs dispersed within the tissue were also shown to be particularly susceptible to tumor formation in mouse epidermis [23] as well as to have large contribution upon wound healing [24], which could also be investigated via our simulation framework. Furthermore, spermatogenic stem cells are also found to be well separated in the planar basal layer in mouse testis, and show a striking similarity to the stem cell arrangement found in our simulations [25]. Other models of the formation of epithelial protrusions and location of stem cells on the tips of these have been studied previously using a particle-based model [26]. By adding substrate deformability and cell type dependent adhesion forces, the model displayed a stable stratified epithelium with stem cells located at protrusions. It will be interesting to combine these models to understand the interplay between our separation mechanism and 3D tissue curvature.

The homogeneous distribution of stem cells arising from our purely mechanical feedback model may complement previous studies [3, 4], in which this was achieved with help of a reversible differentiation scheme or biochemical feedback mechanisms based on diffusible signals. However, in reality, a combination of several of these mechanisms will likely be relevant, also depending on the tissue type. In particular, stochastic cell fate models may be of great interest in the future and could facilitate the description of the plasticity of cells during tissue repair [27, 28]. Also, there exists evidence that more than one SC type maintains some tissues [29,30] or that cell fates are reversible processes [3,31], which could be readily integrated

in our simulation framework. In the presence of local niches, it has also been shown that the ratio of proliferation to random cell movements determines the effective number of stem cells via a mechanism of stochastic competition for space, although this has not been modeled via explicit mechanical simulations [32, 33]. Until now, the vanishing apoptosis rate of SCs and restriction to only asymmetric SC divisions in our model are limiting factors. Allowing for SC loss, or other processes which reduce the SC number, requires symmetric duplication to balance the SC loss to retain their unique identity in the lineage architecture [34]. Furthermore, it would be interesting to see if a purely mechanical model is able to capture the alternation from a quiescent long-lived SC to produce rapidly dividing short-lived progeny during wound repair [35–37].

In this work, we find surprising motility of cells emerging from the interaction of cells with each other. While epithelial cells are generally not motile, motility occurs after epithelial-mesenchymal transformation (EMT) [38], or in some cell lines, in particular MDCK. Active cell migration has been the subject of related work [39–41] and recent findings show that active cell migration does play a role in tissue renewal [42], thus, displaying a promising extension of our work in future studies.

Many aspects of growing tissues have not been addressed in this work. In particular we did not address the role of boundaries and interfaces. Especially in competition setups [8–11, 43] the interface plays a dominant role over the resulting structure. Similarly, three dimensional cell culture is often performed in tissue spheroids, where growth at the surface stabilizes the internal tissue [7, 18, 44, 45] and the dynamics can drive phase separation [46, 47]. A second aspect not addressed here is the rheology or flow response of the tissue. "Analogies between growing dense active matter and soft driven glasses" [48] have been discussed a lot in the literature. In particular, that tissues are generically liquid-like due to cell turnover [6, 49], and whether cell density [50], cell shape [51], or cell-cell adhesion [41] drive a glass like transition on timescales short compared to cell turnover. For a review of further unique properties of self replicating systems like injection of degrees of freedom or giant fluctuations see Ref. [52].

Finally, our results might contribute to a better understanding of the growth and dynamics of cancer. In the cancer stem cell (CSC) hypothesis, it is assumed that malignant tissue follows the same underlying stem cell dynamics. Therapies, which target fast growing tissue, fail, ince the stem cells evade the treatment. A better understanding of the dynamics of CSCs could improve their detection and the design of therapies. Our results suggest that CSCs are isolated and well spread throughout the tumor. To further tackle these questions, it will be necessary to combine our model with models of cellular competition [5, 7–9, 11], and taking the cell cycle explicitly into account might be essential [53, 54]

## Acknowledgments

**Funding information** JE and JK gratefully acknowledge financial support from the Initiative and Networking Fund (IVF) via the grant number ERC-RA-004. Simulations were performed with computing resources granted by RWTH Aachen University under project 'rwth0475'.

**Data accessibility** Source code, simulation data, and analysis scripts will be made publicly available on zenodo.org (10.5281/zenodo.8410957) after acceptance of the manuscript.

# A  Supplementary animations of tissue simulations

Supplementary animations of tissue simulations can be found next to the source code, simulation data, and analysis scripts on *zenodo.org* (10.5281/zenodo.8410957). Uploaded animations are:

- *S1 Video* **Confluent tissue.** Animation of confluent tissue shown in Fig. 1b.

- *S2 Video* **Isolated stem cell.** Animation of isolated SC performing random walk with long persistent segments (see Fig. 2a and sec. D).

- *S3 Video* **Stem cell interaction.** Animation of two SCs interacting with each other via their progeny (see Fig. 2b).

- *S4 Video* **Interaction in Tissue and BD simulations.** Animation of confluent tissue and thermal colloids (see Fig. 3a and Fig. 3b).

# B  Simulation model

The described SC model is implemented in the two-particle growth (2PG) model of Refs. [7,17] and has been adapted in Refs. [9–11].

Each cell is described by two particles, which repel each other via an active growth force

$$F_{ij}^{(G)} = \frac{G}{\left(r_{ij} - r_0\right)^2} \hat{r}_{ij}, \tag{B.1}$$

with unit vector $\hat{r}_{ij}$, distance $r_{ij}$ between the two particles, a constant $r_0$, and growth strength $G$. For non-growing cells, like TD cells, the growth strength is set to zero, $G_{TD} = 0$, and for the sake of simplicity SC and TA cells have the same growth force, $G_{SC} = G_{TA}$.

To prevent overlap of cells, particles of different cells interact via a soft repulsive volume-exclusion force

$$\boldsymbol{F}_{ik}^{(V)} = f_0 \left(\frac{R_{PP}^5}{r_{ik}^5} - 1\right) \hat{r}_{ij}, \quad \text{for} \quad r_{ik} < R_{PP}, \tag{B.2}$$

with exclusion strength $f_0$ and interaction length $R_{PP}$, which sets the length scale of our simulations. Further, cells in contact interact via an attractive adhesion force of the form

$$\boldsymbol{F}_{ik}^{(V)} = -f_1 \hat{r}_{ik}, \quad \text{for} \quad r_{ik} < R_{PP}, \tag{B.3}$$

with adhesion strength $f_1$.

We employ a dissipative particle dynamics-type thermostat [55], with an effective temperature $T$, to account for energy dissipation

$$F_{ij}^{(D)} = \gamma \cdot \left(1 - \frac{dr}{rt}\right)^2 \cdot \left(\boldsymbol{v}_{ij} \cdot \boldsymbol{r}_{ij}\right), \tag{B.4}$$

and random fluctuations

$$F_{ij}^{(R)} = \sigma \xi \cdot \hat{r}_{ij}, \tag{B.5}$$

where $\gamma$ and $\sigma$ are related to fulfill the fluctuation-dissipation theorem [56]. Also, background dissipation is taken into account as

$$\boldsymbol{F}_i^B = -\gamma_b \boldsymbol{v}_i. \tag{B.6}$$

Table 1: Simulation parameters of the standard tissue.

| Parameter | Symbol | Value |
|---|---|---|
| Time step | $\Delta t$ | 0.001 |
| Pair potential interaction range | $R_{PP}$ | 1 |
| Cell expansion constant | $r_0$ | 1 |
| Division distance threshold | $R_{ct}$ | 0.8 |
| New cell particle initial distance | $R_d$ | $10^{-5}$ |
| Growth-force strength | $G$ | 50 |
| Mass | $m$ | 1 |
| Intra-cell dissipation coefficient | $\gamma_c$ | 100 |
| Inter-cell dissipation coefficient | $\gamma_t$ | 50 |
| Background dissipation coefficient | $\gamma_c$ | 0.1 |
| Apoptosis rate | $k_a$ | 0.01 |
| Noise intensity | $k_B T$ | 0.1 |
| Repulsive cell-cell potential coefficient | $f_0$ | 2.39566 |
| Attractive cell-cell potential coefficient | $f_1$ | 3.0 |

A self-consistent velocity-Verlet algorithm [57] is implemented to integrate the equations of motion, and all simulations were performed with periodic boundary conditions.

Cell division is performed, when the two particles of one cell are separated by a critical threshold $R_{ct}$. A new particle is randomly placed near each original particle in a distance $R_d$, and the two new particle pairs form the two daughter cells. Differentiation is implemented at the time of division as described above, and the number of divisions of TA cells is tracked by an internal variable. No mechanism is implemented to prevent (non-growing) TD cells from separating. However, we observe only a marginal number of TD cell divisions in simulations of isolated SCs and SC pairs in free space, and none for confluent tissues. The TD cell size in confluent tissues is always far below the division threshold (see Fig.16), but slightly increases with distance, which is found more pronounced for TD cells generated by SCs in free space (data not shown).

TA and TD cells are stochastically removed from the simulation with a finite apoptosis rate $k_a$. For simplicity, the same rate is chosen for different cell types. Time is measured in terms of the inverse apoptosis rate, and referred to as "generation".

The standard parameter set for our simulations are given in Tab. 1. However, not all of these simulation parameters, have a direct conversion to physical units. As discussed in [17] one has to chose well defined measurable quantities, such as apoptosis rate and range of the pair potential as rescaling units for inverse time and particle diameter, to allow for conversion to physical units and comparison with experiments.

As the number of stem cells in our simulation is fixed by only allowing for asymmetric divisions and no apoptosis, we initialize all simulations with the required number of stem cells, placed at regular distances within the unit box with periodic boundary conditions.

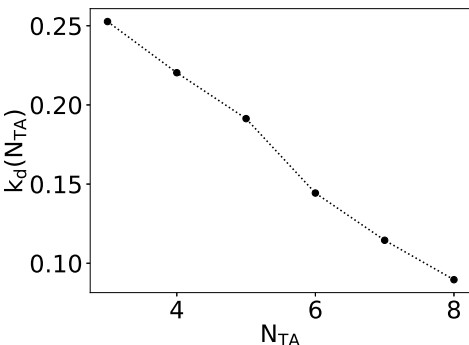

Figure 7: Division rate $k_d$ as function of $N_{TA}$ for proliferating cells in isolated SC simulations.

## C  Division rate of isolated SC

Assuming equal division rates $k_d$ for all types of proliferating cells, we can derive an average cell number from number balance for each cell type

$$n_{SC} = \langle n_{SC} \rangle = \text{const.}$$
$$\langle n_i \rangle = 2^{i-1} \left( \frac{k_d}{k_d + k_a} \right)^i n_{SC},$$
$$\langle n_{TD} \rangle = \left( \frac{2k_d}{k_d + k_a} \right)^{N_{TA}} \frac{k_d}{k_a} n_{SC},$$

(C.1)

where $i$ denotes the TA cell cycle. $k_d$ is obtained from fitting these equations to the cell numbers obtained from simulations of isolated SCs. With increasing $N_{TA}$, the division rate decreases (see Fig. 7).

## D  Stem cell motion

Here, we give some additional insight in the SC motion of isolated SCs and SCs in tissues, which are also discussed in the main text.

An exemplary single stem cell snapshot for $N_{TA} = 4$ is shown, where the SC trajectory over ten generations illustrates the motion of the SC, and accompanied The full trajectory of the SC and center of mass of the respective single stem cell simulation along both directions of the simulation plane (see Fig. 8).

### D.1  Mean squared displacement of isolated stem cells and stem cells in tissues

For active Brownian particles (ABPs) in two-dimensions the mean squared displacement $MSD$ is given as [58,59]

$$MSD = 4\, D_t \Delta t + \frac{v_0^2 \tau_R^2}{2} \left[ \frac{2\Delta t}{\tau_R} + e^{-2\Delta t/\tau_R} - 1 \right],$$

(D.1)

with translational diffusion coefficient $D_t$, characteristic time scale for rotational diffusion $\tau_R$, and propulsion velocity $v_0$. In the short-time limit $dt \ll \tau_R$ the effective diffusion is given by $MSD_{dt \ll \tau_R} \approx 4D_t dt$. For long-time diffusion the effective diffusion is enhanced and the MSD

(a)

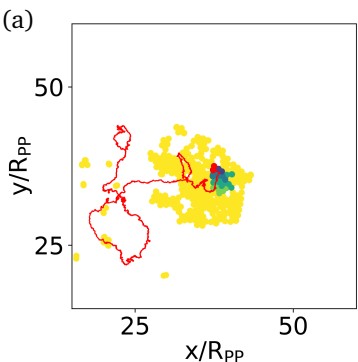

(b)

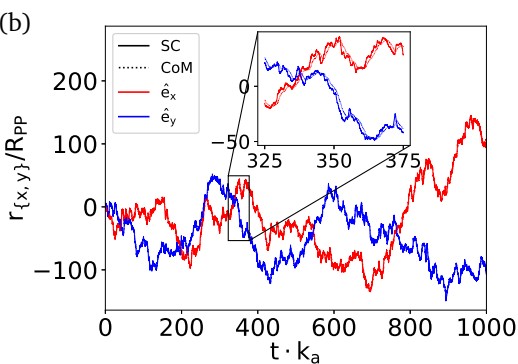

Figure 8: (a) Snapshot of cell population generated by a single SC with trajectory of SC over the ten cell generations (red line). (b) Trajectory of isolated SC (solid line) and center of mass (CoM) of cell population (dotted line) along $\hat{e}_x$ (red) and $\hat{e}_y$ (blue). Inset shows time close-up to highlight how the center of mass follows the stem cell motion. Both, SC and CoM trajectory perform a random walk with surprisingly long persistent segments, a reminiscent of Run'n'Tumble-like motion. Simulation with $N_{TA} = 4$ is shown.

found to be $MSD \approx (4D_t + v_0^2 \tau_R) \, dt$, again linear in time. Between these two regimes, ballistic motion is observed, and non-linear terms enters the MSD, $MSD \approx 4D_t dt + v_0^2 dt^2$.

In the tissue simulations, the stem cells are not ABPs, but move actively due to propulsion by progeny. However, we can quantify the dynamics of the stem cell motion in terms of $D_t$, $\tau_R$, and $v_0$ to gain a deeper understanding. Also different models of self-propelling particles, like ABPs and Run'n'Tumble particles have been shown to be equivalent on larger scales [60, 61], so that we can describe them using the same theory, without making strong assumptions on the underlying propulsion mechanisms.

From the MSDs for simulations of SCs with varying number of TA cell cycles $N_{TA}$ (see Fig. 9), one can see that the displacement does vary little with a change of $N_{TA}$. In particular the short-time diffusion decreases with $N_{TA}$, which could be due to blocking of cell motion at higher density. This resembles the decrease of the translational diffusion coefficient $D_t$ with $N_{TA}$. The long-time diffusion first increases with $N_{TA}$, but then starts to decrease again. Possibly, in increasing number of TA cells around the stem cell initially increases the propulsion force on the stem cell, but saturates at higher number of progeny, which form a more dense halo around the stem cell for more TA cell cycles. The non-monotonic fit parameter for the self propulsion velocity, with maximum at $N_{TA} = 6$, and increasing characteristic rotational diffusion time $\tau_R$ show, that the decrease in propulsion velocity does cause the effective decrease of the long-time diffusion with $N_{TA} > 6$. The characteristic rotational diffusion time $\tau_R$ is found to be in the range from half to more than one apoptosis time, which corresponds to a loss of orientation after times larger than one cell generation.

In tissues, we still can apply the MSD theory to fit to the SC displacement (see Fig. 10). With increasing cell density, controlled by an increase in SC numbers, we find that the translational diffusion coefficient $D_t$ and propulsion velocity $v_0$ decrease, and the SC motion freezes out. The rotational diffusion time is not as strongly effected by the cell density, and approximately half the value compared to the case of an isolated SC, since it is mainly affected by the apoptosis rate.

## D.2 Displacement distribution

To quantify the persistent motion further, we calculate the displacement distribution function and find heavy tails. The heaviness of the tails increases, when the SC is located further away

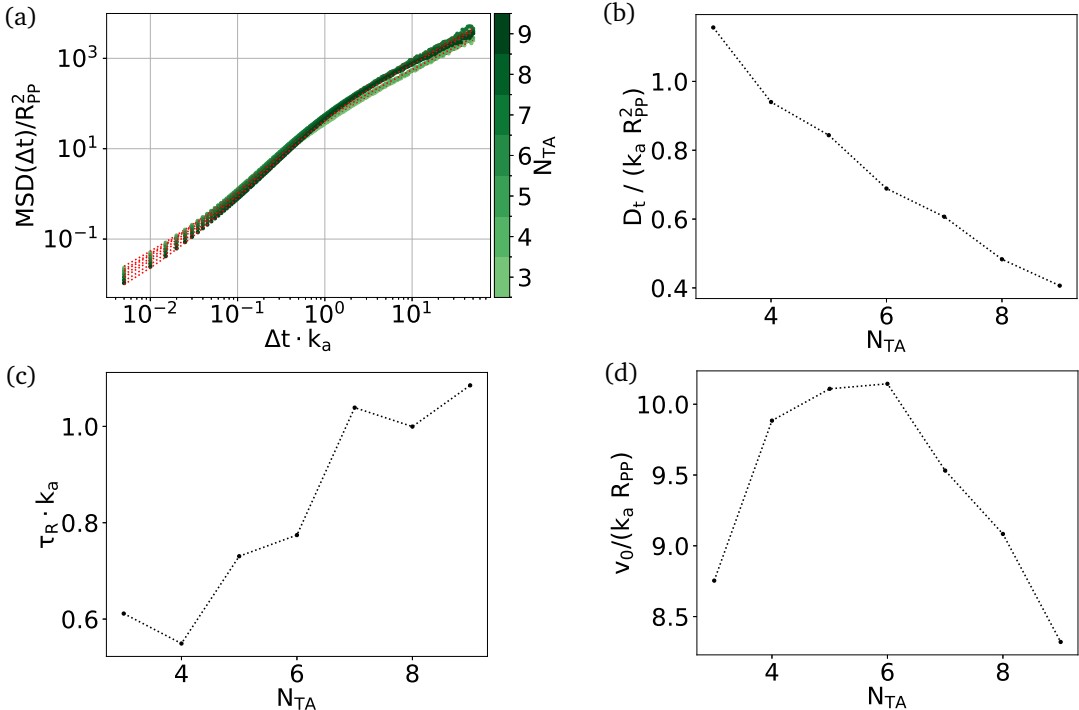

Figure 9: (a) Mean squared displacement for isolated SCs with different $N_{TA}$ (color coded) and fit of ABP model (red dotted lines). (b) Translational diffusion coefficient $D_t$, (c) characteristic rotational diffusion time $\tau_R$, and (d) propulsion velocity $v_0$ obtained from fit function in (a).

from the center of mass in the boundary region of the cell population (see Fig. 11).

## D.3 Overdamped dynamics

The applied self-consistent velocity Verlet integration of the Langevin equations allows for large integration time steps and therefore provides significant faster computations [62]. Effectively, using a finite mass and velocity Verlet integration corresponds to using a higher order integrator with a physically interpretable control parameter, the mass of cells. However, care has to be taken to reproduce overdamped dynamics, which is achieved by setting mass and friction coefficients such that the viscous relaxation time is much shorter than all other relevant timescales, i. e. that viscous drag dominates over inertia even at short timescales. Because inertial effects scale with the mass of the particles, correct reproduction of overdamped dynamics can be verified by showing independence of the results on mass. As an example, we measured the self propulsion of a single stem cell over two orders of magnitude in mass (see Fig. 12). The results are virtually indistinguishable, confirming correct reproduction of overdamped dynamics .

## E   Additional results for simulations of stem cells in tissues

### E.1   Cell densities in confluent tissues

In confluent tissue simulations, we chose the SC number and $N_{TA}$ such that an uninterrupted tissue evolves. Still, the TD cell density around SCs remains very similar to the case of isolated SCs. The TD cells are located closer to the SCs and form a more dense conglomerate. Exemplary cell densities for $N_{TA} = 6$ are shown in Fig. 13a.

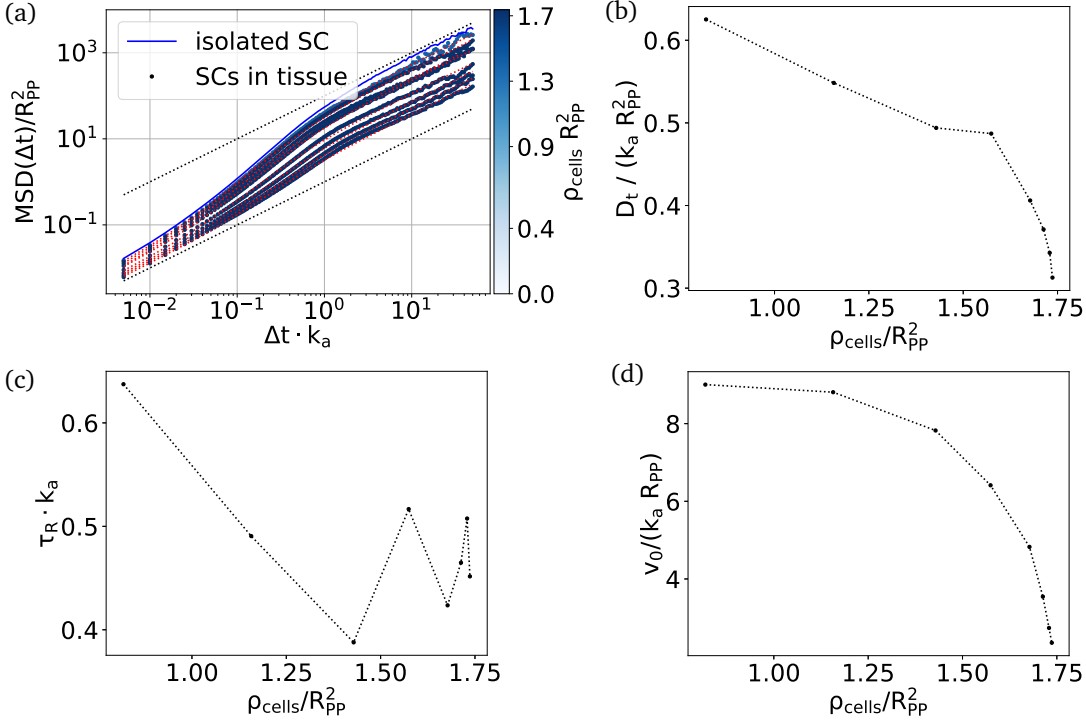

Figure 10: (a) Mean squared displacement for SCs in tissue with different cell density (color coded) controlled by the stem cell number, with $N_{TA} = 6$. (b) Translational diffusion coefficient $D_t$, (c) characteristic rotational diffusion time $\tau_R$, and (d) propulsion velocity $v_0$ obtained from fit function in (a).

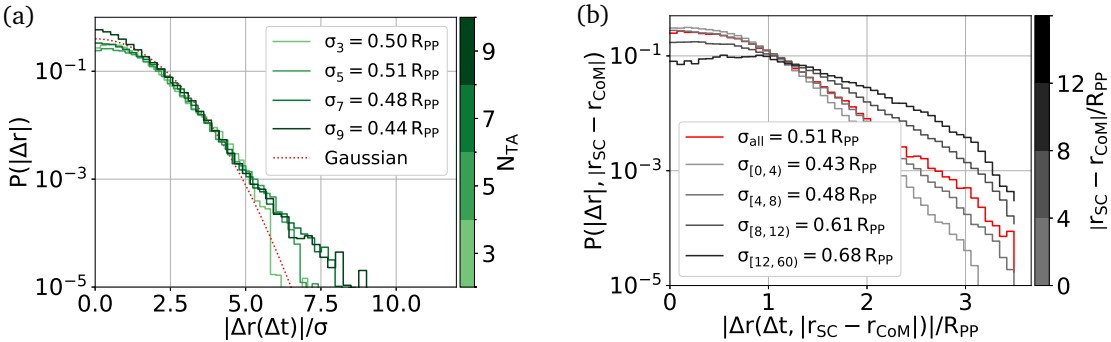

Figure 11: (a) Displacement distribution for an isolated stem cell with different $N_{TA}$ (green color gradient) over normalized displacement. Red dotted line displays normal displacement (Gaussian). Heavy tails emerge from long consistent segments of random walk. (b) Displacement distribution as in (a) for isolated SC with $N_{TA} = 5$, where color gradient (grey) encodes SC-to-CoM distance. With increasing distance heaviness of tails increases. All displacements are calculated for a time step of $\Delta t\, k_a = 0.2$ generations, combining data of multiple long simulation runs.

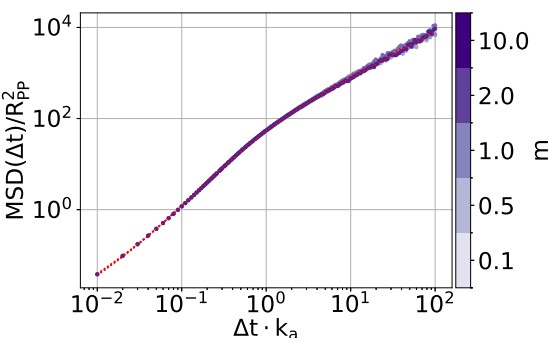

Figure 12: Mean squared displacement for isolated SCs for simulations with different cell masses $m$ (purple color gradient). All simulations are performed with $N_{TA} = 6$.

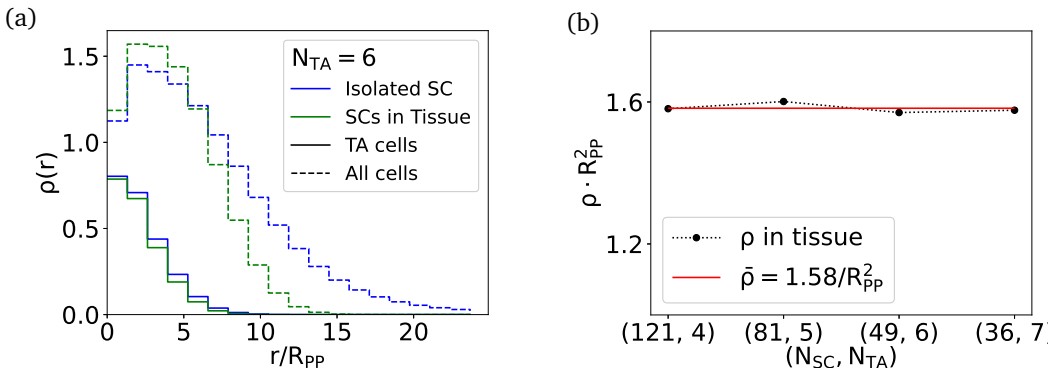

Figure 13: (a) Cell density of TA (solid line) and all (dashed line) cells around the SC for simulations of an isolated SC (blue) and SCs in tissue (green). In tissue, distance is measured to the closest SC. (b) Average cell density of tissue simulations as function of $N_{SC}$, the fixed number of SC, $N_{TA}$ obtained from average cell number over box area.

The total cell density of tissues with different $N_{TA}$ is controlled by the number of SCs in the simulation, and chosen such that they are all very similar (see Fig. 13b).

## E.2 Cluster analysis

To quantify the clustering of SCs in tissues, we calculate the cluster size distribution function is given by

$$\mathcal{N}(n) = \frac{1}{N_{SC}} n \, p(n), \tag{E.1}$$

and represents the fraction of SCs belonging to a cluster of size $n$, where $p(n)$ is the number of clusters of size $n$. The distribution is normalized such that $\sum_{i=1}^{N_S C} = 1$. The average cluster size is given as

$$\langle n \rangle = \frac{\sum_n n p(n)}{\sum_n p(n)}. \tag{E.2}$$

Cells in a distance less than the characteristic TA cell distance estimated from cell numbers belong to the same cluster. Results are shown in Fig. 14.

## E.3 Effective diffusion coefficient and relaxation

The timescale for the thermal colloid simulation in the main text was adjusted by matching the short term diffusion obtained from the MSD of SCs in the tissue simulations. Here, we

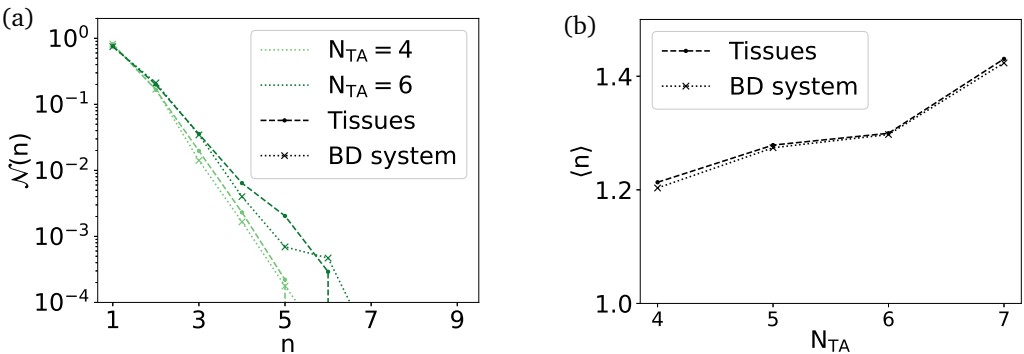

Figure 14: (a) Same as Fig. 3d (b) Average cluster size as function of $N_{TA}$ remains clearly below two SC.

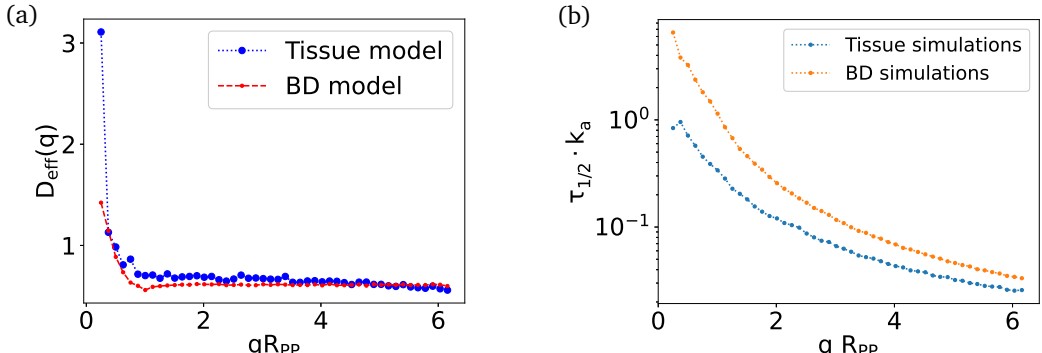

Figure 15: (a) Effective diffusion coefficient $D_{eff}$ measured from structure factor relaxation for $\Delta t = 0.2 k_a$. (b) Relaxation half life time as function of $q$.

calculate the effective diffusion coefficient $D_{eff}$ from the intermediate structure factor. For the relaxation of structures it holds

$$\frac{S(q, \Delta t)}{S(q, 0)} = e^{-D_{eff}(q) \, \Delta t \, q^2 \Delta t}, \tag{E.3}$$

which yields

$$D_{eff}(q) = \log \left\{ \frac{S(q, \Delta t)}{S(q, 0)} \right\} / (q^2 \Delta t). \tag{E.4}$$

Now, we can measure the effective diffusion for short time lags (here: $0.02 k_a$) and see that both models show the same short term diffusivity, as should be (see Fig. 15a).

The relaxation of structures is obtained from the half-life relaxation time of $S(q, t)/S(q, 0)$. We find that this time decreases with $q$, which means that smaller structures relax faster. Further, in the tissue model structures relax faster than in the thermal colloid system and the relaxation speeds up with structure size (see Fig. 15b).

# F Stochastic division rate model

The observed separation of stem cell in tissues is a generic effect arising from short range mechanical interactions between pairs of cells. To highlight this, we replace the size-threshold division (STD) mechanism by a stochastic division with fixed rate $\tilde{k}_d$ (SDR). This turns off the homeostatic pressure control on divisions. We determine the simulation parameter $\tilde{k}_d$ for the SDR model from measurements of $k_d$ in the STD model. Only the case of tissues with $N_{TA} = 5$ is examined.

(a) 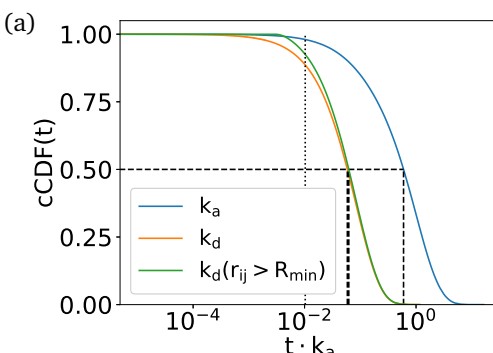 (b) 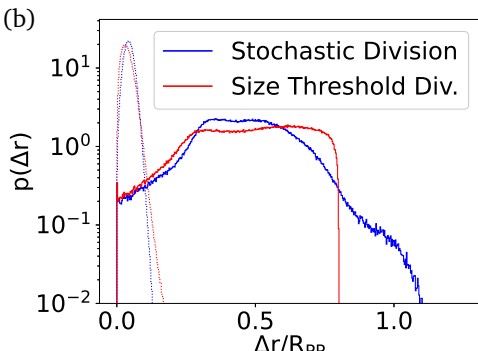

Figure 16: (a) Complementary cumulative distribution function (cCDF) for apoptosis (blue), division events including prevented divisions due to small size (orange), and performed divisions (green). Black dotted line highlights longest time for a prevented division event. Black dashed lines display median of each cCDF. (b) Cellsize distribution for proliferating (SC and TA cells, solid line) and TD cells (dotted lines) in the stochastic division model (blue) and for size threshold divisions (red).

To implement, that cells do not divide at too small size and cause computational flaws, we replace the growth force by a damped harmonic oscillator

$$\boldsymbol{F}_{i,j}^{Gharm.} = -k\boldsymbol{x} - \gamma_k \dot{\boldsymbol{x}} \,, \tag{F.1}$$

where $k = 100$ is the spring constant, and $\gamma_k = 20$ the damping coefficient, chosen in a way to realize critical damping with relaxation time faster than the average division time. The deflection $\boldsymbol{x} = \left(|\boldsymbol{r}_i - \boldsymbol{r}_j| - R_{rest}\right)\hat{\boldsymbol{e}}_{ij}$ is given by the difference between cell size and rest length $R_{rest}$. We set the rest length of proliferating cells, i.e. SC and TA cells, to $R_{rest}^{SC,TA} = R_{ct} = 0.8R_{PP}$ and for TD cells to $R_{rest}^{TD} = 0.1R_{PP}$.

Despite the fact that we tune the oscillator such that cells grow quickly to a adequate size, some cells divide at very small size and cause computational chaos. To prevent this, we introduce a minimum division size, much smaller than the division size threshold used previously. With $R_{min} = 0.15R_{PP}$ we can ensure stable behavior of the computational model after division.

Figure 16a displays the complementary cumulative distribution function (cCDF) of apoptosis, division, and non-permitted division events, i.e. division events at too small cell size. Only a small fraction of events at very short times is permitted and thus, we sufficiently removed the size-threshold division mechanism. Further, the cell size distribution (see Fig. 16b) shows, that TD cells in both models match each other well, and for growing cells, we find that the hard division size threshold changes to a blurred tail. The stochastic division model has equal number of TA cells, both in total and for each generation, and did produce only slightly more TD cells. However, cell numbers fluctuate more, which can be to the detriment of stable tissue homeostasis.

When comparing both models, we find that SCs are still surrounded by their TA cell cloud. In the stochastic division model the TA cell arrangement around the stem cells looks more fringed compared to the deterministic model but no large differences can be found in the measured TA cell distribution (see Fig. 17a, 17b). Still, for both models, the pair correlation function agree very well (see Fig. 17c) and the activity of SC in the tissue remains almost unchanged (see Fig. 17d).

We conclude that the results discussed in the main text are not a consequence of homeostatic pressure growth control, but result from the constant cell outflux generated by the SC. Yet, the pressure feedback on divisions is beneficial to balance cell numbers in the tissue to ensure homeostasis.

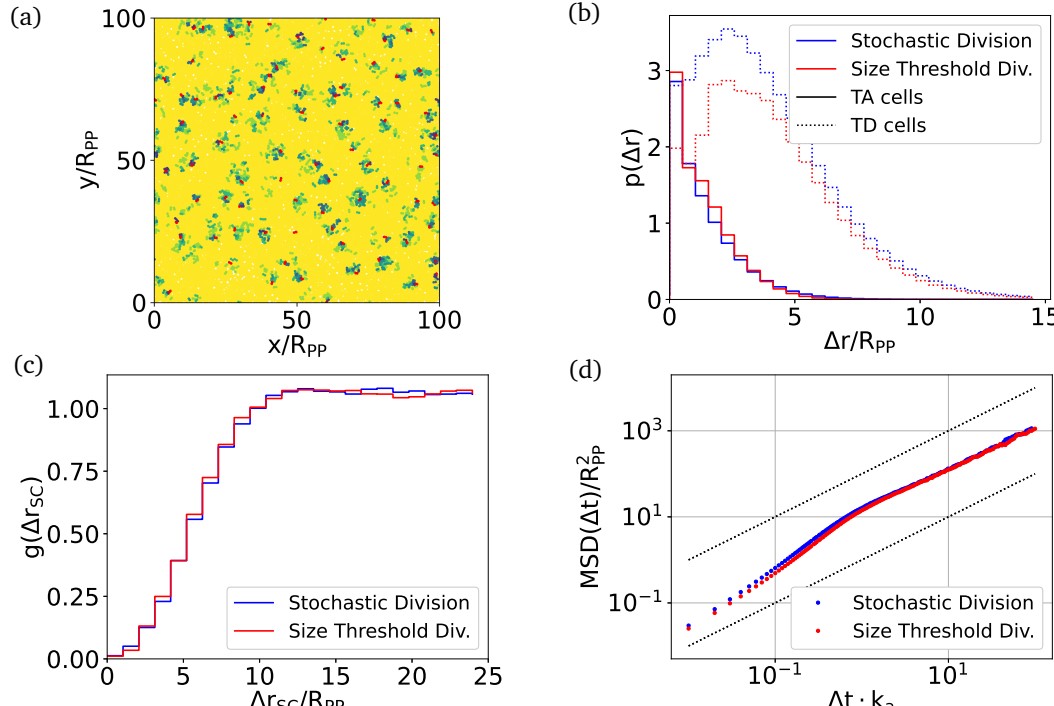

Figure 17: (a) Snapshot for stochastic division model, (b) TA and TD cell distribution around nearest neighbor SC in tissue, (c) pair correlation function, and (d) Mean squared displacement for the respective models. For comparison, stochastic division model is shown by blue lines together with results for model with size threshold division for $N_{TA} = 5$.

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
