# Peer review of "Mechanically-driven Stem Cell Separation in Tissues caused by Proliferating Daughter Cells"

_SciPost Physics, doi:SciPost Phys. 16, 097 (2024)_

## Round 1 · Referee Report · Anonymous (Referee 1) · 2024-2-7

Report

The authors have addressed all my comments sufficiently and I would like to make recommendation for publication without any further revision.

---

## Round 1 · Referee Report · Anonymous (Referee 2) · 2024-2-9

Strengths

Simple model for tissue dynamics with interesting numerical results.

Weaknesses

Perhaps some more citations of relevant work.

Report

I find the paper interesting and easy to read. Regarding the dynamics of dense proliferating tissues, I would suggest to add some refs to the following papers that appear relevant for this work:
-Phys. Rev. Research 2, 043334 – Published 7 December 2020
-Soft Matter, 2020, 16, 5294-5304
-J. Chem. Phys. 153, 201101 (2020)
-Soft Matter, 2017, 13, 3205-3212

---

## Round 1 · Author Response

We thank the reviewer for the valuable comments, which spot weaknesses of the submitted manuscript very well and gave us great hints to improve the quality of this work. We will describe in the following our amendments we made based on the question raised. For the sake of clarity and readability, we repeat each question starting with "Q123...", followed by the respective answer "Answer-Q123...".

Furthermore, we corrected a mistake in Fig. 4(B), where for finite time differences the short time diffsuion matching was incorrect. Now, the correct matching is shown. All results and discussions are not affected by this. Further, few small corrections of typos have been done in the manuscript and are not listed in detail.

Q1. Is there any reason why Newtonian dynamics (rather than overdamped dynamics) is used? (also related to the comment below)

Answer-Q1: This is an excellent comment and we agree with the referee that overdamped dynamics are crucial to describe growing tissues correctly and we should have explained better how this is achieved with Newtonian dynamics, and what their benefits are. Effectively, using a finite mass and velocity Verlet integration corresponds to using a higher order integrator with a physically interpretable control parameter, the mass (Isele-Holder et al., Soft Matter, 2016). However, care has to be taken to reproduce overdamped dynamics, which is achieved by setting mass and friction coefficients such that the viscous relaxation time is much shorter than all other relevant timescales, i.e. that viscous drag dominates over inertia even at short timescales. Because inertial effects scale with the mass of the particles, correct reproduction of overdamped dynamics can be verified by showing independence of the results on mass. As an example, we measured the self propulsion of a single stem cell (see comment Q2 below) in a range of two orders of magnitue in mass. The results are virtually indistinguishable, confirming correct reproduction of overdamped dynamics. In the revised manuscript, we include a discussion on this topic including a plot of the MSD for various masses as an appendix (D.3) and refer to this in the methods (Sec. 2).

Q2. The mechanism for effective propulsion is not explained clearly. I am not sure if this propulsion is due to inertial effects, in which case the crossover timescale from ballistic to diffusive regime will correspond to the inertial timescale of the particles/cells. I think the authors should repeat the measurements with overdamped dynamics to rule out that the effective propulsion is not due to the inertial regime of the dynamics.

Answer-Q2: To verify that the propulsion is not due to inertial effects, we plot the MSD of a stem cell for various masses (new appendix subsection D.3). The independence of MSD from mass demonstrates that it is not due to inertial effects. We are convinced that the propulsion of stem cells is due to the do the growth of the progeny due to the significant differences between Fig.9 and Fig.12. We added this aspect in appendix D.3 and refer to it in the manuscript.

Q3. Many models of epithelial tissues, on the other hand, do include self-propulsion, since most epithelial cells are also motile. Perhaps comment on this.

Answer-Q3: We agree with the referee, in fact, biological systems provide much more rich phenomena then the ones included in this model, and indeed we have incorporated cellular motility in the two particle growth model (e.g. Basan et al. PNAS 2013, Marel et al. NJP 2014, Garcia et al. PNAS 2015). Two reasons stopped us from doing so in this work: - We want to demonstrate that the text-book version of renewal by stem cells is enough to explain stem-cell separation if one considers mechanics – without invoking further mechanisms like cellular motility. - Epithelial cells are typically (not diseased, not cell line) non motile. Motility appears after the so called EMT (Epithelial to mesenchymal transition), or in some cell lines (in particular MDCK). However, there are now indications that indeed progenitors actively migrate out of the stem cell niche in the intestinal epithelium. This aspect is added to the discussion as a promising direction for future research.

Q4. In the model the apoptosis rate of the SC cells is zero. Did you assume the apoptosis timescale for SC to be much longer than the simulation timescales?

Answer-Q4: Indeed, we set the apoptosis rate for SC to zero to explore the dynamics the stem cells experience due to its progeny, and not worry about the dynamics in stem cell number. As the referee points out, this can be seen as assuming the apoptosis timescale of SCs (and timescale between symmetric SC divisions) to be much longer than the simulation timescales. In reality of course, SCs are occasionally lost, and do divide symmetrically (duplicate) as well. How a stable Stem cell population is maintained is subject of current research (Jörg et al, Annual Review of Condensed Matter Physics 2021, Greulich et al, PNAS 2016, Greulich et al, Development 2021), and we will incorporate SC death and symmetric divisions in future works. In the manuscript, we added this aspect to the outlook.

Q5. Does the cell division conserve the center of mass?

Answer-Q5: Yes and no: - Yes, the center of mass of the cell is conserved: At the time of a division, we place two new particles very close to each old particle of the dividing cell. This displacement vector of the new particles is chosen at random, but very short (1.25e-5 times the division length $R_{ct}$, or $1e-5R_{PP}$). - No, the center of mass of the cell colony is not conserved: During cell division, and apoptosis, the total mass is naturally not conserved. I.e. a division event slightly shifts the center of mass towards the event. To capture mass conservation, the interstitial fluid has to be modeled as well, which is usually not done in tissue simulations. See e.g. Ranft et al. EPJE 2012 for a discussion of the physics of such a two fluid model. Estimates therein suggest that effects of the permeation flow of the interstitial fluid only play a role on lengthscales much larger than the system studied here.

Q6. Page 4 paragraph 2: What is N_{TA}? is it the maximal number of division cycles?

Answer-Q6: Yes, N_{TA} is the maximal number of division cycles transient amplifying cells can undergo and was introduced in the introduction on page 2. To improve readability and clarity, we again mention this in Sec.3.

Q7. Page 4 paragraph 2: Note that k_d is not fixed in our simulation model, but controlled by pressure. What is the functional dependence of k_d on pressure?

Answer-Q7: One key feature of our simulations is, that cells divide when reaching a division size threshold R_{ct}. Their growth dynamics depend on inter- and intracellular mechanical forces. Hence, in our simulations, k_d decreases exponentially with pressure (see e.g. Podewitz et al. EPL 2015), the corresponding growth rates (k = k_d - k_a) agree well with experimental data (Montel et al. PRL 2011, Podewitz et al. EPL 2015). In this work, we focus on the homeostatic state, where on average cell death is balanced by cell loss. In this case, we typically expand k linearly around the homeostatic pressure (Basan et al. HFSP 2009). For this work however, the key point is that pressure is an intrinsic quantity, i.e. homogeneous across the system. Thus, the cells adapt their growth rate globally to cell loss.
We added citations to the manuscript at this point to improve contextualization of the manuscript.

Q8. Figure 2,3,4: are all the data taken at steady state?

Answer-Q8: Yes, we do only take into account data in steady state, i.e. after reasonably long time after initialization. To clarify this in the manuscript, we add this statement in the figure caption.

Q9. In the model: The cell is composed of two particles. It seems that the only interaction between these two particles within the same cell is the active growth force, Eq. (2). In the case of TD cells, where growth force is zero, how do you prevent the two particles from separating?

Answer-Q9: Just to clarify: Besides the growth force, the particles of the same cell also interact by internal friction (\gamma_{internal}) and the corresponding noise. However, we do not prevent the TD cell particles from separating. On very rare occasions, TD cells can separate far enough, and thus divide. However, as one can see in the appendix in Fig. 16b (note: Fig.15b in old version), the cell size of TD cells in tissues is much smaller than the division cutoff. We only observe a marginal fraction of TD cell divisions for the case of isolated SCs and pairs of SCs in free space, but not in confluent tissues. We mention this now briefly in appendix B.

Q10. In the model: What is the initial state of the two particles when a new cell is born?

Answer-Q10: After each division, the position of the two particles of the old cell remains unchanged. The two new particles, which are required to create new cells, are placed at very short distances (1e-5 the interaction length) with random orientation next to the old particles. We explain this now in appendix B.

Q11. Table 1: what do these numbers correspond to in physical units?

Answer-Q11: Connecting simulation units with physical units is a challenging task. This goes back to the Major Comment 1: mass is just a control parameter of the integration. Thus directly converting to SI units does not yield proper numbers. We thus generally use non-dimensional quantities, where corresponding quantities are related to each other. Time and length scales for example are measured in units of apoptosis rate and interaction length, corresponding to “cell generations” and “cell size”. In Basan et al, Physical Biology (2011), we fitted our simulation results to results on growing tumor spheroids, and details about parameter fitting and conversion to SI units can be found therein. A comment and citation to this reference has been added in appendix B.

---

## Round 1 · List of Changes

• Update of Fig. 4(b)
  • Add discussion on overdamped dynamics and measurement of the MSD of a single SC over cell mass in appendix D.3 and Fig. 12.
  • Add reference to App.D.3 in Sec.2 and App.B
  • Add discussion on cell motility in Sec.8
  • Add discussion of SC removal (e.g. apoptosis, differentiation, or delamination) to Sec.8
  • Add clarification of definition of N_{TA} and citations regarding mechanosensitivity (k_d) in Sec.3
  • Amend main text figure captions to clarify that data is taken in steady-state
  • Add discussion on TD cell separation and initial cell state and comment on conversion to physical units of the simulation parameters and citations in App.B
  • Few minor corrections of typos and spelling mistakes.

---

## Round 2 · Author Response

We would like to express our sincere thanks to the editor for his efforts to obtain a further peer review to assess the quality of the work. We also thank the reviewer for his efforts and are very pleased with the positive report.

We are happy that we can further improve the contextualization of the work through his suggestions and have now taken them up in the discussion.

---

## Round 2 · List of Changes

• Extend the discussion regarding possible extensions of the model, in particular spheroids and glass-like vs fluid behavior, based on the suggestions of the referee, and included additional references. (one paragraph in the discussion section)
  • Update one existing reference, which was cited as a preprint, and now has been accepted and published.

---

## Editorial Decision

published